# Mothers' satisfaction with health extension services and the associated factors in Gamo Goffa zone, Southern Ethiopia

Mark Mergia[1⊚], Mulugeta Shegaze[2⊚], Gistane Ayele[2⊚], Eshetu Andarge[2⊚], Yordanos Gizachew Yeshitila[3⊚]*, Biresaw Wassihun[4‡], Abayneh Tunje[2‡], Peter Memiah[5‡]

1 Gamo Zone Health Department, Arba Minch University, Arba Minch, Southern Ethiopia, Ethiopia, 2 School of Public Health, College of Medicine and Health Sciences, Arba Minch University, Arba Minch, Ethiopia, 3 School of Nursing, College of Medicine and Health Sciences, Arba Minch University, Arba Minch, Ethiopia, 4 Department of Midwifery, College of Medicine and Health Sciences, Arba Minch University, Arba Minch, Ethiopia, 5 Center for International Health, Education, and Biosecurity, Institute of Human Virology, University of Maryland School of Medicine, Baltimore, Maryland, United States of America

⊚ These authors contributed equally to this work.
‡ These authors also contributed equally to this work.
* yordanos.gizachew@yahoo.com

**Data Availability Statement:** All relevant data are within the manuscript and its Supporting Information files.

**Funding:** The author(s) received no specific funding for this work.

## Abstract

### Background

Providing compassionate and respectful maternity services in order to bring satisfaction to service users is one of the health extension services packages in Ethiopia. Though the service utilization and its associated factors have been well documented so far, yet there is a dearth of information on client satisfaction; particularly among rural women who are mostly the target groups of health extension services within the Ethiopia. Thus, this study aimed to assess mothers' satisfaction with health extension services and the associated factors in the Gamo Goffa Zone, of Southern Ethiopia.

### Methods

A community based cross sectional study was conducted among 478 women from February 1st to February 28th of 2018. A pre-tested and interviewer administered questionnaire was used to collect the data. The study participants were selected using a systematic random sampling technique by allocating a proportion to each rural kebeles. Both bivariate and multivariate logistic regression analysis were performed to identify associated factors. Odds ratio with its 95% confidence interval was used to determine the degree of association. Level of statistical significance was set at a p-value of less than 0.05.

### Result

The overall proportion of mothers who were satisfied with health extension services was 37.4% with 95% CI (33–44%). Mothers who had good family monthly income [AOR = 4.66 (95% CI: 1.1, 20.6)], whose husbands participated in the health extension program [AOR = 4.02 (95% CI: 2.0, 8.1)], who had recent participation in planning of health extension

**Competing interests:** The authors have declared that no competing interests exist.

programs [AOR = 5.75 (95% CI: 2.0, 16.5)], who were recognized as a model family [AOR = 2.23 (95% CI: 1.1, 4.6)] had higher odds of satisfaction with health extension services.

## Conclusion

Satisfaction with the health extension services was low among women in the study area. A better household monthly income, women's and their husbands' participation in health extension services and being a member of a model family were positively associated with their satisfaction. Interventions to enhance women's satisfaction in the health extension service need to focus on household-based involvement of them and their husbands in planning and implementing the services. Likewise, expansion of experiences of model families in the community would raise satisfaction levels.

## Introduction

Client satisfaction is one of the excellent ways that enables health programs to assess the impact of their services. Satisfied clients are likely to come back for the services they need and to recommend the services to others [1,2]. Maternal satisfaction refers to the mother's positive evaluation of distinct dimensions of health care [3,4]. More than 85 percent of Ethiopians reside in rural areas which are difficult to access by the health professionals. Compared to the global average, Ethiopia has several poor key health outcome indicators. Rates of death attributed to preventable and avoidable causes of diseases is still high in Ethiopia. A considerable number of children and mothers are still dying due to failures in reaching them with high impact interventions [5, 6].

Health Extension Program (HEP) is a ground-breaking community-based program that was started in Ethiopia in 2003 as a response of the Alma- Ata conference [7]. Under an enormous countrywide drive, over 38,000 rural Health Extension Workers (HEWs) have been deployed over 15,000 health posts (HPs) (with a catchment area population of 5,000 on average) [8]. The HEWs are providing basic health services to mothers and members of the community in rural area for facilitating closer contact between the health workers and the communities [9, 10]. The program involves encouraging women's participation in decision-making processes and promotes community empowerment [11,12]. Its philosophy is based on transferring the right knowledge and skills to the households so that they can take the responsibility for realizing and keeping their own health. [13]. In the last decade, incredible gains have been registered in terms of improving access and utilization of maternal services such as increasing contraceptive acceptance rate, focused antenatal care (FANC), skilled birth attendance, improved health seeking behavior, expanding vaccination services, malaria control and prevention, and reduction of HIV infections [14,5].

As per studies, a positive relationship between clients and providers is central to any health service delivery system. Thus, a low satisfaction of mothers implies a gap between current experiences and their expectation from the services that would finally lead them to move away to distant public health facilities or costly private health facilities by the essence of seeking for quality healthcare services [15–17]. In Africa, clients´ satisfaction with health care services is one of the most important factors determining the utilization of the services. So, assessment of mothers' satisfaction with health extension service is crucial to measure maternal service utilization [18–21].

It has been suggested that health extension services improve the overall health care system in Ethiopia. Despite the expansion in the implementation of preventive and curative health services in Ethiopia, the maternal mortality ratio is still high at 412 out of 100,000 live births in 2016 [6]. Majority of the HEP packages are reproductive and maternal health services and are implemented with mothers' as their primary targets. However, the satisfaction level of mothers to the health extension service (HES) is not yet adequately studied in the country. The studies conducted in relation to the HEP primarily focused on its implementation challenges, successes and impact [22, 23,24]. To the best of our knowledge, there was one study in Jimma zone which employed a mixed method study in 2013[17]. In line with the focus of compassionate and respectful care in the current health sector transformation plan of the country [25] and the expected variation across regions of the country in the level of satisfaction in general and HES in particular, the findings of this study is timely and would contribute to the local programme planning and policy making at large. Moreover, the study has covered a large population of mothers at zonal level. Therefore, this study was aimed at assessing mothers' satisfaction with health extension service and the factors associated with it in Gamo Goffa zone, Southern Ethiopia.

## Materials and methods

### Setting and duration of the study

The study was conducted in the Gamo Goffa Zone, which was one of the 14 zones in Southern Nations, Nationalities and Peoples Region (SNNPR), Ethiopia. The zone covers a total area of 12,581.4 square kilometers consisting 15 woredas and 2 town administrations. It has 482 kebeles (448 rural and 34 urban). According to 2007 Ethiopian Fiscal year census projection, the estimated number of total populations of the study area in 2010 E.C was 2,089,527; of which 1,023,868 (49%) were males and 1,065,659 (51%) were females. Among the existing population, 13.08% of them were living in urban areas. The estimated number of women in reproductive age (15–49) were 475,546, of which 413,326 were dweller of rural kebeles. A total of 416,525 households (an average of 4.9 persons per household) within the zone and 362,027 of them were rural households. Regarding the current health facility distribution, the zone has 2 general hospitals, 5 primary hospitals, 75 health centers and 471 HPs. A total of 713 rural health extension workers were serving the rural community with ratio of 1 HEW to 2456 people [26]. As of 2019, the Gamo Goffa zone is administratively divided in to two separate zones namely Gamo and Goffa zones. The study was conducted from February 1st to February 28th of 2018.

### Study design and population

A community based cross-sectional study was conducted among mothers of reproductive age (15–49 years) who reside in rural kebeles of the Gamo Goffa zone.

### Inclusion and exclusion criteria

Mothers who lived for greater than 6 months in a kebele with functional HP and who were served by the HEP were included in the study. Those mothers who were critically ill, mentally disabled and who were not able to respond to the interview questions for any reason were excluded from the study.

## Sample size and sampling procedure

A single population proportion formula was used to estimate the sample size required for the study. The sample size calculation assumed the proportion (p), the estimated level of satisfaction of mothers 83% [17], 95% confidence level, margin of error of 5% and design effect of 2 which gave a sample size of 434. In consideration of a 10% non-response rate, the final sample size was 478 mothers. Of all woredas of the study area, four (Melo Koza, Zalla, Boreda, and Arba Minch Zuria) were selected using lottery method of simple random sampling. The determination of samples from each kebele was made proportionally upon the numbers of households. Finally, systematic random sampling was employed to select the study participants from each kebele.

## Data collection instruments and procedures

Data were collected using a structured questionnaire developed after a thorough review of documents, guidelines, and manuals related to HEPs, and previous studies conducted in rural areas. The research team and a team of experts working in the health extension programme checked the contents and relevance of the questions by emphasizing on the dimensions of satisfaction. This was done to ensure face and content validation of the items in the questionnaire. The questionnaire was initially prepared in English and translated into the local language and —then back translated to English to check for consistency. A face-to-face interview was conducted by data collectors who completed 10th grade and were dwellers of the respective kebeles. Supervisors were recruited from the nearby health centers in order to oversee the data collection process. Households were contacted by the assistance of guiders from the local women development army members. When women were not available at their home during the first visit, an attempt was made to reach them twice again. By far, those with no opportunity to reach them at the third trial were considered as non-respondents and; a woman next door was interviewed. During data collection, a reliability analysis was done on 162 cases (greater than or equal to 10-fold of the 16 items) and the result showed a good score of internal consistency between the items (Cronbach's alpha = 0.89).

## Definitions and measurements

**Health extension workers.** Health care practioners deployed at HP level primarily to provide promotive and preventive health care services recruited based on nationally agreed criteria that include residence in the village, capacity to speak local language, graduation from 10th grade, and willingness to remain in the village and serve the local community [27].

**Model families.** Households that are trained in some of the components of the HE packages and able to implement these packages and influence their relatives and neighbors to adopt the same practices [27].

**Health development army.** The health development army (HDA) refers to an organized movement of communities designed to improve the implementation capacity of the health sector by engaging communities to identify local challenges and corresponding strategies. Each functional team comprise up to 30 households and is further divided into smaller groups of six members, commonly referred to as one-to-five networks [27].

**Health post.** A primary health care unit at the grass-root level of the health care delivery system of Ethiopia staffed with HEWs and serving 3000–5000 populations [27].

Mothers' satisfaction with different aspects of health extension service was assessed with dimensions on convenience of the service, satisfaction with courtesy (a polite speech or action) of HEWs, and satisfaction with quality of service provision using 16 satisfaction items that have five-point Likert scales. The responses, 'very satisfied' and 'satisfied' were coded and regarded as "satisfied"; whereas, responses 'very dissatisfied', 'dissatisfied' and 'neutral' were regarded as "unsatisfied".

Overall satisfaction was measured as individuals score 75% of the response of the sixteen satisfaction items were categorized as "satisfied"; while those who scored less than 75% of the items were categorized as "unsatisfied".

## Data processing and analysis

Data completion and consistency were manually checked. Incomplete and inconsistent reports were discarded. The data were initially coded, checked and entered in Epi-Data software version 3.1. Then it was exported to statistical package for social sciences (SPSS) software version 23.0 for cleaning and statistical analysis. Descriptive statistics such as frequencies, mean and standard deviation was carried out to see the distribution of the participants within the variables included in the study. After cross-tabulation of each explanatory variables with the outcome variable and checking the fulfillment of chi-square assumption, binary logistic regression analysis model was fitted.

Initially, bivariate logistic regression analysis was carried out to select variables for multivariable analysis. Variables with p value <0.25 in the bivariate analysis were selected as candidates for multivariable logistic regression analysis model. Model fitness was checked using Hosmer and Lemeshow goodness of fitness test. Multicollinearity among the explanatory variables was checked using Variance Inflation Factor (VIF>10). To control for possible confounding and identify independent factors associated with mothers' satisfaction with HES, a final multivariable model was built using backward stepwise method. Level of statistical significance was declared at a p-value of less than or equal to 0.05. Adjusted odds ratio with its 95% CI was used to show the strength of association between each explanatory variable and the outcome variable. The results of the analysis were presented in descriptive texts and tables.

## Data quality management

Data collectors and supervisors were provided with a daylong intensive training on the techniques of data collection and components of the instrument. Prior to the commencement of the data collection, pretest was conducted among 24 mothers (5% of the sample size) which were not included in the study. Based on the findings of the pre-test, ambiguous questions were amended. An ongoing formative checkup for completeness and consistency of responses was made by the supervisors on a daily basis.

## Ethics consideration and consent to participate

Ethics clearance was issued from the Institutional Review Board of Arba Minch University (Number: CMHS/9441/54 on 04/04/2010 Ethiopian Calandar). Oral consent was obtained from each participant. Written consent was not sought owing to the fact that majority of rural women in our setup are not able to read. Before obtaining oral consent from each participant, a letter of support and approval for undertaking the research in the local setup was obtained from the government officials in each kebele. This was done by the approval of the IRB since there was no sensitive information regarding the data. Privacy as well as confidentiality of participants was asserted. In any case, their right to withdraw from the study at any time was assured.

## Results

### Socio-demographic characteristics of the study participants

All of the mothers (100%) were participated in the study. The mean age of participants was 31.58 (SD = ±7.25). Of all, 25.3% were in the age group of 25–29 years. More than half (55.9%)

of the participants were followers of Orthodox Christianity and nearly half (51.5%) of them were from Goffa ethnic group. Regarding women's occupation, 87.2% of them were house-wives. A majority of the study participants (66.1%) were not able to read and write. About half of the participants (46.9%) earn 15–30 USD per month (Table 1).

**Table 1. Socio demographic characteristics of the study participants in Gamo Goffa Zone, Southern Ethiopia, 2018 (N = 478).**

| Variables | Frequency | Percent |
|---|---|---|
| **Age in a year** | | |
| **15–19** | 12 | 2.5 |
| **20–24** | 72 | 15.1 |
| **25–29** | 121 | 25.3 |
| **30–34** | 92 | 19.2 |
| **35–39** | 104 | 21.8 |
| **40–44** | 54 | 11.3 |
| **45–49** | 23 | 4.8 |
| **Religion** | | |
| **Orthodox** | 267 | 55.9 |
| **Protestant** | 211 | 44.1 |
| **Ethnicity** | | |
| **Amhara** | 5 | 1.0 |
| **Gamo** | 211 | 44.1 |
| **Goffa** | 246 | 51.5 |
| **Wolayta** | 16 | 3.3 |
| **Marital status** | | |
| **Married** | 452 | 94.6 |
| **Separated** | 12 | 2.5 |
| **Divorced** | 7 | 1.5 |
| **Widow** | 7 | 1.5 |
| **Educational status** | | |
| **Cannot read and write** | 316 | 66.1 |
| **Only read and write** | 80 | 16.7 |
| **Primary level** | 49 | 10.3 |
| **Secondary level** | 33 | 6.9 |
| **Occupation** | | |
| **House wife** | 417 | 87.2 |
| **Merchant** | 61 | 12.8 |
| **Family monthly income (USD)** | | |
| **<15** | 33 | 6.9 |
| **15–30** | 224 | 46.9 |
| **31–45** | 71 | 14.9 |
| **46–60** | 108 | 22.6 |
| **61–90** | 34 | 7.1 |
| **>90** | 8 | 1.7 |
| **Family size** | | |
| **<4** | 55 | 11.5 |
| **4–5** | 184 | 38.5 |
| **6–7** | 188 | 39.3 |
| **>7** | 51 | 10.7 |

**Table 2.  Mothers' interaction with health extension services in Gamo Goffa Zone, Southern Ethiopia, 2018 (N = 478).**

| Variable | Category | Frequency | Percent |
|---|---|---|---|
| **Source of information about HES** | HEWs | 384 | 80 |
| | Health professionals | 51 | 11 |
| | Mass media | 26 | 5 |
| | Neighbor | 17 | 4 |
| **Visit to the HP (in the last one year)** | Yes | 415 | 86.8 |
| | No | 63 | 13.2 |
| **Frequency of HP visit (in the last one year)** | 1 times | 194 | 47 |
| | 2 times | 133 | 32 |
| | 3 times | 63 | 15 |
| | 4 and above | 25 | 6 |
| **Travel time to HP (in minutes)** | ≤30 minutes | 274 | 57.3 |
| | 30–60 minutes | 163 | 34.1 |
| | ≥ 60 minutes | 41 | 8.6 |
| | Mean± SD = 33±23 | | |
| **Availability of HEWs on job at HP** | Always | 358 | 86 |
| | Occasional | 40 | 10 |
| | Rarely | 17 | 4 |
| **Received service from HEWs** | No | 76 | 15.9 |
| | Yes | 402 | 84.1 |
| **Visited by HEW (in the last one year)** | No | 69 | 14.4 |
| | Yes | 409 | 85.6 |
| **Frequency of Visit by HEW(N = 409)** | 1 times | 180 | 44 |
| | 2 times | 121 | 30 |
| | 3 times | 80 | 20 |
| | ≥ four times | 28 | 7 |
| **Husband involved in HE services (in the last one year)** | No | 345 | 72.2 |
| | Yes | 133 | 27.8 |
| **Mothers participated in the planning of HE activities (in the last year)** | No | 217 | 45.4 |
| | Yes | 261 | 54.6 |
| **Mothers participated in HDA** | No | 234 | 49.0 |
| | Yes | 244 | 51.0 |
| **Mothers recognized as a model family** | No | 301 | 63.0 |
| | Yes | 177 | 37.0 |
| **Mothers know the model families in the kebele (N = 301)** | No | 130 | 43 |
| | Yes | 171 | 57 |
| **Place of residence for HEWs** | HP | 405 | 84.7 |
| | Town | 16 | 3.3 |
| | With family | 57 | 11.9 |
| **Perceive that the HES is adequate** | No | 357 | 74.7 |
| | Yes | 121 | 25.3 |

## Mothers' interaction with health extension services

The data revealed that all the study participants had information about HES. HEWs were the source of information for the services provided for 80% of mothers in the study area. Majority of mothers (86.8%) visited HP in the last one year. More than a third of mothers (37%) were recognized as model families (Table 2).

**Table 3. Mothers' exposure to health extension services in Gamo Goffa Zone, Southern Ethiopia, 2018 (N = 478).**

| Type of service received | Frequency | Percent |
|---|---|---|
| **Immunization** | 357 | 89 |
| **Excreta disposal** | 243 | 60 |
| **Malaria** | 238 | 59 |
| **Family planning** | 228 | 57 |
| **Nutrition** | 213 | 53 |
| **HIV/AIDS, other STDs** | 187 | 47 |
| **TB** | 174 | 43 |
| **Personal hygiene** | 165 | 41 |
| **Healthy house environment** | 137 | 34 |
| **Food supply and safety measures** | 120 | 30 |
| **Antenatal care** | 116 | 29 |
| **First aid** | 116 | 29 |
| **Solid and liquid waste disposal** | 107 | 27 |
| **Water supply and safety measures** | 104 | 26 |
| **Insect and rodent control** | 80 | 20 |

## Utilization of health extension services

Eighty-nine and sixty percent of the participants received services on immunization and waste management respectively. Programs such as skilled birth attendance, insecticide and rodent control, and water supply and safety were given the least attention in the community (Table 3).

## Mothers' level of satisfaction with health extension services

Of all the participants, 37% were satisfied with the HES they received (scored 75% and above in the composite score from the 16 items). Satisfaction of the participants in the specific aspects of the services received are also presented below (Table 4).

## Factors associated with mothers' satisfaction with health extension service

Monthly income of households, the availability of HEW at HP, involvement of husband in the activities of HEWs, mothers' participation in planning of HEWs activities, participated in HDAs in the previous year and being a model family were significant factors associated with mothers' overall satisfaction.

Participants whose monthly income was greater than 60 USD were 5.3 times more likely to be satisfied with HES [AOR = 5.3 with 95%CI (1.1, 25.2)] than those with monthly income less than 15 USD. The study participants whose husbands were participated in HES were 4.02 times more likely to be satisfied with HES [AOR = 4.02 with 95% CI (2.0, 8.1)] compared with those whose husbands did not participate. Mothers who participated in the planning of HEWs activities the year prior to the study were also 5.75 times more likely to be satisfied [AOR = 5.75, 95% CI (2.0, 16.5)] compared with those who did not participate in planning the activities of HEWs. Mothers participated in the activities of the HDAs in the previous year were 3.42 times more likely to be satisfied [AOR = 3.42, 95% CI (1.3, 9.2)] than those who did not participate in the activity of HDAs (Table 5).

## Discussion

This study was aimed to assess the level of mothers' satisfaction with the HES and to identify independent factors associated with their satisfaction in Gamo Goffa zone, southern Ethiopia.

**Table 4. Mothers' level of satisfaction with different aspects of health extension services in Gamo Goffa Zone, Southern Ethiopia, 2018 (N = 478).**

| S.Num | Variables | Satisfied | Dissatisfied |
|---|---|---|---|
| | | Num(%) | Num (%) |
| 1 | Simplicity and trouble-free service | 220(46) | 258(54) |
| 2 | Availability of needed instruments | 212(44) | 266(56) |
| 3 | Facilitation of referral for consultation | 207(43) | 271(57) |
| 4 | Friendliness and Courteous of HEWs | 208(44) | 270(56) |
| 5 | Provision of attention for clients | 196(41) | 282(59) |
| 6 | Provide appropriate time for examination or counseling | 196(41) | 282(59) |
| 7 | Keeping privacy of clients | 207(43) | 271(57) |
| 8 | General satisfaction with HES | 240(50) | 238(50) |
| 9 | Presence of variety of services | 222(46) | 256(54) |
| 10 | Quality of the service | 217(45) | 261(55) |
| 11 | Extent of met need | 226(47) | 252(53) |
| 12 | Availability and access to information | 210(44) | 268(56) |
| 13 | Perceived benefit from service received | 208(44) | 270(56) |
| 14 | Recommendation of the service for a friend/neighbor in need | 205(43) | 273(57) |
| 15 | Satisfaction with specific services during home visit | 212(44) | 266(56) |
| 16 | Comfort with follow-up service | 203(42) | 275(58) |
| | Overall satisfaction | 299(63) | 179(37) |

The finding from the study revealed that the overall proportion of mothers who were satisfied with HES was 37.4% with 95% CI (33, 44%). This value is very low compared with findings of a similar study conducted in Jimma zone, 83% [17]. This discrepancy may be due to difference in knowledge of mothers, cultural diversity, and the techniques used to compute overall satisfaction. This study measured satisfaction in a more comprehensive way considering additional aspects of the HES. Though the low satisfaction might be related to the measurement, it implies that satisfaction with the HES is far lower in the study area seen in light of the aspects considered.

Family monthly income of mothers was significantly associated with their satisfaction with HES. This finding is consistent with the study conducted in West Gojjam Zone [28]. In studies from clinical care setup and with a paid service, monthly income was inversely associated with satisfaction [29] or else did not show any association with satisfaction to the service [30–32]. Women from a higher family income might be more educated, autonomous in household decisions and health care choices and would also have better alternatives in accessing the curative services provided in other higher-level institutions. Even though HES is free of charge in the health care delivery of the country, households with lower income might not be satisfied with the service owing to its least focus on curative services which majorly incurs cost to them at higher health facilities. Unmet needs for curative services was also concerns of households from previous studies in Ethiopia [17,33–35].

Consistent with the findings from the Jimma study [17], husbands' involvement in HES is also another factor associated with satisfaction of mothers in the service. Male involvement has resulted in improved uptake of reproductive and maternal services in previous studies in the country [36–40]. Husbands who were involved in HES are more likely to be committed in approving women's initiation to receive the service and act as a model to the community. Moreover, they might be more professionals and competent, of recognized status by the community and in a position to confront challenging work conditions and share the information they gained from these experiences to their wives [17,41]. Thus, these would have resulted in a

**Table 5. Factors associated with mothers' satisfaction with health extension services in Gamo Goffa Zone, Southern Ethiopia.**

| Variable | | Dissatisfied | Satisfied | COR (95%. I)P. Value | AOR (95% C.I), P-Value |
|---|---|---|---|---|---|
| **Household monthly income (USD)** | <15 | 28 | 5 | 1.00 | 1.00 |
| | 15–30 | 144 | 80 | 3.11 (1.2, 8.4) *.025 | 3.34 (.8, 13.9) .094 |
| | 31–45 | 42 | 29 | 3.87 (1.3, 11.2) * .013 | 5.30 (1.1, 25.2) .036** |
| | 46–60 | 61 | 47 | 4.31 (1.5, 12.0) * .005 | 4.66 (1.1, 20.6) .042** |
| | 61–90 | 19 | 15 | 4.42 (1.4, 14.2) * .013 | 3.64 (.6, 20.5) .143 |
| | >90 | 5 | 3 | 3.36 (0.6, 18.7) .167 | 2.41 (.2, 30.1) .495 |
| **Frequency of HP visit in the previous year** | 1 time | 123 | 71 | 1.00 | 1.00 |
| | 2 times | 68 | 65 | 1.66 (1.1, 2.6) * .027 | .98(.5, 2.0) .945 |
| | 3 times | 36 | 27 | 1.30 (0.7, 2.3) * .375 | 1.33 (.5, 3.7) .590 |
| | 4 and above | 11 | 14 | 2.20 (0.9, 5.1) * .066 | 1.42 (.4, 5.2) .594 |
| **Time taken to reach at HP on foot (in minute)** | 1–30 minutes | 147 | 127 | 1.00 | 1.00 |
| | 31–60 minutes | 115 | 48 | 0.48 (0.3, 0.7) * .001 | .68 (.3, 1.3) .268 |
| | >61minutes | 37 | 4 | 0.13 (0.0, 0.4) * .000 | 1.23 (.2, 6.7) .810 |
| **The availability of HEW on job at HP** | Always | 193 | 165 | 1.00 | 1.00 |
| | Occasional | 31 | 9 | 0.34 (0.2, 0.7) * .006 | .15 (.1, .4) .000** |
| | Rarely | 14 | 3 | 0.25 (0.1, 0.9) * .032 | .56 (.1, 3.1) .499 |
| **Received the service from health extension at HP** | No | 72 | 4 | 1.00 | 1.00 |
| | Yes | 227 | 175 | 13.88 (5.0, 38.7) * .000 | 2.93 (.3, 27.7) .349 |
| **Frequency of visit of HHs by HEWs** | 1 times | 124 | 56 | 1.00 | 1.00 |
| | 2 times | 64 | 57 | 1.97 (1.2, 3.2) * .005 | .79 (.4, 1.7) .551 |
| | 3 times | 30 | 50 | 3.69 (2.1, 6.4) * .000 | 1.26 (.5, 3.0) .600 |
| | 4 and above | 13 | 15 | 2.55 (1.1, 5.7) *.023 | 1.57 (.5, 5.3) .464 |
| **Involvement of husband in HES** | No | 269 | 76 | 1.00 | 1.00 |
| | Yes | 30 | 103 | 12.15 (7.5, 19.6) * .000 | 4.02 (2.0, 8.1) .000** |
| **Mothers participated in planning of HES** | No | 205 | 12 | 1.00 | 1.00 |
| | Yes | 94 | 167 | 30.35 (16.1, 57.2) *.000 | 5.75 (2.0, 16.5) .001** |
| **Mothers participated in activity of HDA** | No | 218 | 16 | 1.00 | 1.00 |
| | Yes | 81 | 163 | 27.42 (15.5, 48.6) * .000 | 3.42 (1.3, 9.2) .015** |
| **Mothers recognized as a model family** | No | 255 | 46 | 1.00 | 1.00 |
| | Yes | 44 | 133 | 16.76 (10.5, 26.6) * .000 | 2.23 (1.1, 4.6) .028** |

*-significant at a p-value less than 0.25

**-significant at a p-value less than 0.05

better satisfaction of women in the study. This implies that a sustained effort is on demand from the stakeholders to strengthen participation of husband's in the HES.

Mothers' participation in the HDA was also associated factor of mother's satisfaction with HES. This is plausible since HDAs are networks of the community who work with sub-team 1 to 5 networks of households in the community [27], participation in their activities does mean participation to the HES in general. A study from Australia suggested that community participation will result in higher community satisfaction with health services as well as better health outcomes [42]. In our case, as mothers are more exposed to the activities by the HDAs, they will more likely be satisfied with the HES as HDAs are volunteers who are implementers of the HES [27].

In the present study, mothers who were recognized as a model family were more likely to be satisfied compared with mothers who were not recognized as a model family. A consistent finding was reported from previous related studies from West Gojam [16] and Jimma zones

[17], North West and South West Ethiopia respectively. Model families are the early adopters of desirable health practices as a role model to implement the HE packages and their selection is used as a strategy to improve household behaviors. They are deemed to have acceptance and credibility by the community and are presumed to diffuse health messages so that the desired practices and behaviours can be easily adopted by the rest of the community[9]. Since the model families are at the front seat and interested in the HES, they are more likely to be satisfied than those who are not. This suggests the expansion of model family coverage in the local setup and by extension in the country.

So, the stakeholders should work to address the identified factors particularly graduating households as a model family that would increase their participation and satisfaction in the HES leading to improvement in basic health services utilization. Husband involvement also needs a reconsideration.

This study was conducted among rural mothers from a large population of Gamo Goffa Zone and has considered a wider dimension of satisfaction to the HES. However, it has few limitations to consider. Family monthly income was measured as an average income for the household which may not always give a reliable response as majority of rural women in the study setup were not educated and therefore may not be able to estimate their monthly income in term of cash. Besides this, mother's evaluation of their own income should have also been sought and it would have been better if a household wealth index was created. Also, the scale used to measure satisfaction in this study was not rigorously validated. Moreover, the study looked to women's satisfaction from their own perspective. The perspectives and perceptios of HEWs, and others working for the improvement of the HEP is the area which needs a further evidence from a qualitative study.

## Conclusion

The study revealed that satisfaction to HES is very low in Gamo Goffa Zone. Family average monthly income, involvement of husband in HES, mothers' participation in planning of HES and in the activities of HDAs in the previous year and mothers' recognition as a model family were significant factors associated with their overall satisfaction.

## Supporting information

**S1 File. Revised manuscript with track changes.**
(DOCX)

**S2 File. Response to reviews.**
(DOCX)

## Acknowledgments

The authors are grateful for the data collectors and the study participants for their voluntariness in the data collection process and Arba Minch University for providing ethical clearance.

## Author Contributions

**Conceptualization:** Mark Mergia, Mulugeta Shegaze, Gistane Ayele.

**Data curation:** Mark Mergia, Mulugeta Shegaze, Gistane Ayele, Eshetu Andarge, Yordanos Gizachew Yeshitila, Biresaw Wassihun, Abayneh Tunje.

**Formal analysis:** Mark Mergia, Mulugeta Shegaze, Gistane Ayele.

**Funding acquisition:** Mark Mergia, Mulugeta Shegaze, Gistane Ayele.

**Investigation:** Mark Mergia, Mulugeta Shegaze, Gistane Ayele, Eshetu Andarge, Yordanos Gizachew Yeshitila.

**Methodology:** Mark Mergia, Mulugeta Shegaze, Gistane Ayele, Eshetu Andarge, Yordanos Gizachew Yeshitila, Biresaw Wassihun, Abayneh Tunje.

**Project administration:** Mark Mergia, Mulugeta Shegaze, Gistane Ayele.

**Resources:** Mark Mergia, Mulugeta Shegaze, Gistane Ayele, Yordanos Gizachew Yeshitila, Biresaw Wassihun, Abayneh Tunje.

**Software:** Mark Mergia, Mulugeta Shegaze, Gistane Ayele, Eshetu Andarge, Yordanos Gizachew Yeshitila, Biresaw Wassihun, Abayneh Tunje.

**Supervision:** Mark Mergia, Mulugeta Shegaze, Gistane Ayele, Eshetu Andarge, Yordanos Gizachew Yeshitila, Biresaw Wassihun, Abayneh Tunje.

**Validation:** Mark Mergia, Mulugeta Shegaze, Gistane Ayele, Eshetu Andarge, Yordanos Gizachew Yeshitila.

**Visualization:** Mark Mergia, Mulugeta Shegaze, Gistane Ayele, Eshetu Andarge, Yordanos Gizachew Yeshitila, Biresaw Wassihun, Abayneh Tunje, Peter Memiah.

**Writing – original draft:** Mark Mergia, Mulugeta Shegaze, Gistane Ayele, Eshetu Andarge, Yordanos Gizachew Yeshitila.

**Writing – review & editing:** Mark Mergia, Mulugeta Shegaze, Gistane Ayele, Eshetu Andarge, Yordanos Gizachew Yeshitila, Biresaw Wassihun, Abayneh Tunje, Peter Memiah.

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
