## [Decision Letter · Decision Letter 0]

27 Feb 2020

PONE-D-19-34888

Mothers’ satisfaction with health extension services and the associated factors in Gamo Gofa Zone, Southern Ethiopia

PLOS ONE

Dear Mrs. Yeshitila,

Thank you for submitting your manuscript to PLOS ONE. After careful consideration, we feel that it has merit but does not fully meet PLOS ONE’s publication criteria as it currently stands. Therefore, we invite you to submit a revised version of the manuscript that addresses the points raised during the review process.

We would appreciate receiving your revised manuscript by Apr 12 2020 11:59PM. To enhance the reproducibility of your results, we recommend that if applicable you deposit your laboratory protocols in protocols.io, where a protocol can be assigned its own identifier (DOI) such that it can be cited independently in the future. For instructions see: http://journals.plos.org/plosone/s/submission-guidelines#loc-laboratory-protocols

We look forward to receiving your revised manuscript.

Kind regards,

Nülüfer Erbil, Ph.D, Prof.

Academic Editor

PLOS ONE

Journal Requirements:

2. Please include additional information regarding the survey or questionnaire used in the study and ensure that you have provided sufficient details that others could replicate the analyses. For instance, if you developed a questionnaire as part of this study and it is not under a copyright more restrictive than CC-BY, please include a copy, in both the original language and English, as Supporting Information.  If the original language is written in non-Latin characters, for example Amharic, Chinese, or Korean, please use a file format that ensures these characters are visible.

3. Please state whether you validated the questionnaire prior to testing on study participants. Please provide details regarding the validation group within the methods section.

4. Please amend your current ethics statement to address the following concerns: Please explain why was written consent was not obtained, how you recorded/documented participant consent, and if the ethics committees/IRBs approved this consent procedure.

Reviewers' comments:

Reviewer's Responses to Questions

**Comments to the Author**

1. Is the manuscript technically sound, and do the data support the conclusions?

Reviewer #1: Yes

Reviewer #2: Yes

Reviewer #3: Partly

2. Has the statistical analysis been performed appropriately and rigorously? 

Reviewer #1: Yes

Reviewer #2: Yes

Reviewer #3: Yes

3. Have the authors made all data underlying the findings in their manuscript fully available?

Reviewer #1: Yes

Reviewer #2: Yes

Reviewer #3: No

4. Is the manuscript presented in an intelligible fashion and written in standard English?

Reviewer #1: Yes

Reviewer #2: Yes

Reviewer #3: No

5. Review Comments to the Author

Reviewer #1: Mothers’ satisfaction with health extension services and the associated factors in Gamo Gofa Zone, Southern Ethiopia

Thank you for inviting me to review the above-mentioned manuscript. Please see my comments as follow:

1.Abstract: Authors should report the odds of mother that had lower satisfaction not mothers with higher satisfaction.

Methods

1.Please mention the name of country along with the region of study.

2.Please state the inclusion/exclusion criteria of the study.

3.The validity and reliability of the questionnaire that authors used in this study did not confirm properly.

Results

1.Table 1: for better understanding of economic situation, please exchange the currency of your country to the US Dollar. It was much better, if you have asked about how participants evaluate their economic situation? For instance good, poor, moderate. May you please consider this matter in the limitations of study.

2.Table 1: please provide the mean and SD for continuous data such as age.

3.Please provide some midwifery information such as, gravida, para, number of abortion, and the previous mode of delivery…

4.What authors mean about model family?

5.Table 4: please put the number and percentages together.

6.The availability of HEW on job at HP was significant and seems that occasional availability was a source of dissatisfaction, please explain.

Discussion

1.Please write the objective of the study at the beginning of the discussion.

2.Please mention the limitations of the study at the end of discussion.

3.Authors found that only small percentages of mothers are satisfied with the services received from health providers. It might be due to the different instrument that they used in this study and also did not validate it.

Reviewer #2: This cross sectional study which looked at Mothers’ satisfaction with health extension services and the associated factors in Gamo Gofa Zone, Southern Ethiopia is commendable and will add to the already existing body of knowledge on this topic. The title of the study is appropriate, the sample size determination and methodology were systematic. The study investigates important issues related to maternal satisfaction in a low income country. The subject is relevant for reflections on prevention of maternal deaths in the country. Over all the manuscript clearly states factors which contribute maternal satisfaction in relation to health extension service provision which intern may contribute to decrease maternal mortality in the country, I suggest the paper is suitable for publication once after the author come with necessary correction and hope that editors will consider publishing the article.

Besides that I have some points to mention:

-some of grammatical errors should be corrected (subject to verb agreement); for example data is the plural form so it is better to use “were” instead of “was” and see it for “kebeles” as well

-At abstract: you state compassionate and respectful maternity services; it doesn’t go with your objectives, so the authors should see it again. In addition the word “Methods” are repeated.

-Background is too long and is better to exchange first and second paragraph

-In method section p12 L 124; you have used design effect; why it is important to use it since you didn’t clearly state weather you have used cluster sampling or not

-Under the Results, the labelling of the tables had the title of the whole study repeated under each table. My suggestion would be that Table 1 should be titled: Sociodemographic characteristics of the respondents, Table 2 should be: Mothers’ interaction with health extension services, and so on. Again in table 1under age category it is mentioned as “30-3” see it again. When you try to mention study subjects as ‘participants’ in some part and ‘respondents’ I suggest you to use consistent way of writing.

-Table 5 is not clear. There should be a key to explain the single and double asterix.

-In discussion section it was stated as this discrepancy may be due to difference in knowledge respondents, cultural diversity, satisfaction items and the techniques used to compute overall satisfaction. This study measured satisfaction in a more comprehensive way considering additional aspects of the HES. My biggest concern is the difference in satisfaction items how it can be different as it was Likert scales. Please explain briefly

- better to add declaration section in your manuscript

Reviewer #3: Thank you for the opportunity to review this manuscript under title the Mothers’ satisfaction with health extension services and the associated factors in Gamo Gofa Zone, Southern Ethiopia.

Title:

It's better to capitalize the first letter in each word in the title.

At the beginning of each paragraph leaves a space and begins writing.

Abstract:

1-In line 34 delete repeated word (method).

2-in line 41 review the results and style of writing. For example, focuses on the [AOR=4.02 (95% CI 45 2.0, 8.1),

3-Not use Abbreviations in the abstract. In line 55. Write done Health Extension Workers.

Method:

In general, the method needs revision and rearmament of the subtitle. For example:

1-If you agree change the (Study Area and period) to Setting and Duration of the Study. In line 104.

2- Clarify the estimated number of total population in line (108) under title population estimated or Population with estimation of sample size.

3-Explain inclusion and exclusion criteria.

Results:

1-Revision all tables.

2-Table number 2 not clear. Unifying the form of writing numbers and the symbol %.

6. PLOS authors have the option to publish the peer review history of their article (what does this mean?). If published, this will include your full peer review and any attached files.

Reviewer #1: Yes: Parvin Abedi

Reviewer #2: No

Reviewer #3: Yes: Warda Hassan Abdullah

---

## [Author Response · Author response to Decision Letter 0]

18 Mar 2020

RESPONSE TO REVIEWS BY EDITOR AND REVIEWERS

PONE-D-19-34888

Mothers’ satisfaction with health extension services and the associated factors in Gamo Goffa Zone, Southern Ethiopia

Editor’s comments

Journal Requirements: When submitting your revision, we need you to address these additional requirements.

Response: We have made the revised manuscript to conform to plosone’s style requirements.

2. Please include additional information regarding the survey or questionnaire used in the study and ensure that you have provided sufficient details that others could replicate the analyses. For instance, if you developed a questionnaire as part of this study and it is not under a copyright more restrictive than CC-BY, please include a copy, in both the original language and English, as Supporting Information. If the original language is written in non-Latin characters, for example Amharic, Chinese, or Korean, please use a file format that ensures these characters are visible.

Response: We have included a copy of the questionnaire used in this study as a “supporting information”. 

3. Please state whether you validated the questionnaire prior to testing on study participants. Please provide details regarding the validation group within the methods section. 

Response: We really appreciate for your concern on the gap in information with regard to the validity and reliability of the survey instrument. We admit that we have not followed the steps for scale validation in its strict sense. However, we have done things related to face and content validations as the research team discussed on dimensions of satisfaction and the clarity and relevance of the survey questions and reached agreement before testing the questions on the study women. The research team (experts of public health, nursing and midwifery) and a team of local experts working with in the health extension programme were participated in suggesting the components of satisfaction and appropriate wording of questions. Right before data collection, a pre-test was made on 5% (24 mothers) to see the appropriateness of the questions to the mothers in charge and the necessary amendments were made to make the questions suitable to the survey. Moreover, concurrent to the data collection, a reliability analysis was made to check for internal consistency on 162 samples of women (Cronbach’s alpha=0.89). We apologize for not going through the other validation steps which are not commonly done in our country. An additional statement has been provided in the “manuscript with track changes” document.

4. Please amend your current ethics statement to address the following concerns: Please explain why was written consent was not obtained, how you recorded/documented participant consent, and if the ethics committees/IRBs approved this consent procedure.

Response: Thank you for your concern and remarks on this important area. Oral consent was obtained owing to the fact that majority of the rural women in our setup are not able to read and write except few who might have attended a formal/ an informal education or else working in the public/private sector. Before going to obtain oral consent from each participant, a letter of support and approval for undertaking the research in the local setup was obtained from government officials in each kebele. The IRB approved this procedure initially at proposal defense stage for there was no sensitive issue regarding the research task. These statements have been incorporated to the “revised manuscript with track changes” file in the “methods” section.

 

REVIEWERS' COMMENTS

Reviewer #1: 

Mothers’ satisfaction with health extension services and the associated factors in Gamo Gofa Zone, Southern Ethiopia

Thank you for inviting me to review the above-mentioned manuscript. Please see my comments as follow:

1.Abstract: Authors should report the odds of mother that had lower satisfaction not mothers with higher satisfaction.

Response: Our categorization of the outcome variable was initially as “satisfied=1” and “dissatisfied=0”. Thus, the success category would be satisfaction and the odds ratio we found from this analysis were >1 (shows a positive association). Therefore, we have reported in favor of higher satisfaction owing to these reasons -the odds ratio is positive and the success variable initially coded as 1 in the logistic regression model was “satisfied” and as 0 was “dissatisfied”.

Methods 

1.Please mention the name of country along with the region of study.

Response: We have included it in the “methods” section (line 106).

2.Please state the inclusion/exclusion criteria of the study.

Response: We have included it in the “methods” section (L 121-126). 

3.The validity and reliability of the questionnaire that authors used in this study did not confirm properly.

Response: We really appreciate your concern on the gap in information with regard to the validity and reliability of the survey instrument. We admit that we have not followed the steps for scale validation in its rigorous sense. However, we have done things related to face validation and content validations as the research team discussed on dimensions of satisfaction and the clarity and relevance of the survey questions and reached agreement before testing the questions on the study women. The research team (experts of public health, nursing and midwifery) and a team of local experts working with in the health extension programme were participated in suggesting the components of satisfaction and appropriate wording of questions. Right before data collection, a pre-test was made on 5% (24 mothers) to see the suitability of the questions to the mothers in charge and the necessary amendments were made to make the questions suitable to the survey. Moreover, concurrent to the data collection, a reliability analysis was made to check for internal consistency on 162 samples of women (Cronbach’s alpha=0.89). Forgive us for not going through the other validation steps which are not commonly done in our country. An additional statement has been provided in the “manuscript with track changes” document.

Results

1.Table 1: for better understanding of economic situation, please exchange the currency of your country to the US Dollar. 

Response: We have corrected it to USD in table 1 of the revised manuscript.

2. It was much better, if you have asked about how participants evaluate their economic situation?

 For instance-good, poor, moderate. May you please consider this matter in the limitations of the study.

Response: We have included it as a limitation in the “discussion” section of the revised manuscript.

3.Table 1: Please provide the mean and SD for continuous data such as age.

Response: Dear reviewer, the mean and standard deviation of age was already presented in the text under “socio-demographic characteristics of the study participants”. 

3. Please provide some midwifery information such as, gravida, para, number of abortions, and the previous mode of delivery…

Response: Dear reviewer, we do not have these data from the outset. Since we did not get any theoretical or literature ground that supports as these factors were associated with satisfaction to a given program, we did not include them as a variable/explanatory variable in this study. Had these variables were included in the model, the probability that they are too much correlated to maternal age or educational status is high. However, if we had the data now, we would have described it here as a “background” characteristic.

5.What authors mean about model family?

Response: This has been defined in the “definitions and measurement” section as – “Model families: Households that are trained in some of the components of the HE packages and able to implement these packages and influence their relatives and neighbors to adopt the same practices [27]”. Kindly look to it.

6.Table 4: please put the number and percentages together.

Response: We have revised it accordingly in the revised manuscript.

7.The availability of HEW on job at HP was significant and seems that occasional availability was a source of dissatisfaction, please explain.

Response: This is just related to mother’s satisfaction with the service directly goes with the usual presence or occasional presence or absence of HEWs in the health post. Health extension workers are providers of the service. When women come to get the service they wanted , they will be happy or satisfied when they get the service providers in the respective health posts, most importantly at all times.

Discussion

1.Please write the objective of the study at the beginning of the discussion.

Response: Thank you for your directions. We have included our study objectives at the beginning of the “discussion” section.

2.Please mention the limitations of the study at the end of discussion.

Response: Thank you again for your direction here too. We have incorporated the limitations of the study at the end of the discussion.

3.Authors found that only small percentages of mothers are satisfied with the services received from health providers. It might be due to the different instrument that they used in this study and also did not validate it.

Response: We sincerely apologize for the tool validation which should have been done in a rigorous way before we go for data collection. However, we have tried to show what steps we ensue with the tool development (face and content validation), pre-tested the questionnaire on 5% of the sample of women and conducted a reliability analysis on the first 162 mother’s data (Cronbach’s alpha=0.89). We admit the assertion on the small percentage of satisfaction and had already discussed on it in the “discussion” section…...” Though the low satisfaction might be related to the measurement, it implies that satisfaction with the HES is far lower in the study area seen in light of the aspects considered” …. (lines 264-265). We have included a limitation statement in the “discussion” section.

 

Reviewer #2: 

This cross-sectional study which looked at Mothers’ satisfaction with health extension services and the associated factors in Gamo Gofa Zone, Southern Ethiopia is commendable and will add to the already existing body of knowledge on this topic. The title of the study is appropriate, the sample size determination and methodology were systematic. The study investigates important issues related to maternal satisfaction in a low -income country. The subject is relevant for reflections on prevention of maternal deaths in the country. 

Over all the manuscript clearly states factors which contribute maternal satisfaction in relation to health extension service provision which in turn may contribute to decrease maternal mortality in the country, I suggest the paper is suitable for publication once after the author come with necessary correction and hope that editors will consider publishing the article. 

Response: Thank you very much for your inspiring remarks.

Besides that, I have some points to mention:

1. Some of grammatical errors should be corrected (subject to verb agreement); for example, data is the plural form so it is better to use “were” instead of “was” and see it for “kebeles” as well

Response: Thank you for your corrections. We have made the necessary corrections where appropriate.

2. At abstract: you state compassionate and respectful maternity services; it doesn’t go with your objectives, so the authors should see it again. 

Response: Compassionate and respectful care has been mentioned in the “abstract” and the “background” section for its close connection with satisfaction to service and its strategic connection with the current Ethiopian government health sector transformation plan and the SDGs.

3. In addition, the word “Methods” are repeated. 

Response: Thank you for your meticulous observation. We have removed one of the “methods” and the “abstract” section.

4. Background is too long and is better to exchange first and second paragraph

Response: We have exchanged the first and the second paragraphs and therefore references number 1-14 have been exchanged accordingly. 

5. In method section p12 L 124; you have used design effect; why it is important to use it since you didn’t clearly state weather you have used cluster sampling or not.

Response: Since the study is at zonal level, we have employed multi- stages sampling where the woredas and kebeles were considered as clusters. We have selected some woredas and kebeles to represent the remaining woredas and kebeles because we considered that these woredas and kebeles were externally homogenous with regard to satisfaction to HE services.

6. Under the Results, the labeling of the tables had the title of the whole study repeated under each table. My suggestion would be that Table 1 should be titled: Sociodemographic characteristics of the respondents, Table 2 should be: Mothers’ interaction with health extension services, and so on. 

Response: We have written the label in such a way just to conform to the principle in writing titles of tables and figures as it should be self-explanatory- things like where, what, and when should be answered.

7. Again, in table 1under age category it is mentioned as “30-3” see it again. 

 Response: Thank you very much for your meticulous observation. We have corrected it under “table 1”.

8. When you try to mention study subjects as ‘participants’ in some part and ‘respondents’ I suggest you to use consistent way of writing. 

Response: Thank you for your direction here. We have made it to be consistent as “participants”.

9. Table 5 is not clear. There should be a key to explain the single and double Asterix.

Response: Thank you for the correction. We have clarified further about the symbols.

10. In the “discussion section” it was stated as this discrepancy may be due to difference in knowledge of respondents, cultural diversity, satisfaction items and the techniques used to compute overall satisfaction. This study measured satisfaction in a more comprehensive way considering additional aspects of the HES. My biggest concern is the difference in satisfaction items how it can be different as it was Likert scales. Please explain briefly

Response: Thank you for expressing your concern. We found it a confusing expression and removed it from the stated section. 

11. It is better to add declaration section in your manuscript.

Response: We did not add a declaration section as part of the manuscript owing to its absence in the “plos one manuscript formatting”. However, almost all the contents known to be part of “declaration” in the other reputable journals are there in the Plos one online submission system and we have provided the appropriate response. 

 

Reviewer #3: 

Thank you for the opportunity to review this manuscript under title the Mothers’ satisfaction with health extension services and the associated factors in Gamo Gofa Zone, Southern Ethiopia.

Title:

1. It's better to capitalize the first letter in each word in the title.

Response: We have followed the plose one formatting style that the title should be in sentence case.

2. At the beginning of each paragraph leaves a space and begins writing.

Response: Thank you for your recommendation. We have corrected accordingly.

Abstract:

1-In line 34 delete repeated word (method).

Response: Thank you very much for your meticulous observation to improve our work. We have removed the repeated word.

2-In line 41 review the results and style of writing. For example, focuses on the [AOR=4.02 (95% CI 45 2.0, 8.1),

Response: We ask for forgiveness that we could not get the remark here. 

3-Not use Abbreviations in the abstract. In line 55. Write done Health Extension Workers.

Response: Thank you for your suggestion. We have made the necessary correction.

Method:

In general, the method needs revision and rearmament of the subtitle. For example:

1-If you agree change the (Study Area and period) to Setting and Duration of the Study. In line 104.

Response: We have corrected accordingly.

2- Clarify the estimated number of total populations in line (108) under title population estimated or Population with estimation of sample size.

Response: The estimated number of women of reproductive age is estimated from the total population by using a conversion factor for Southern Nations, Nationalities and Peoples Region (SNNPR) of Ethiopia. The conversion factor is derived from average percentages of women 15-49 during the most recent census in the region. Sample size is calculated by considering these women of reproductive age as a source population and using the usual assumptions for single population proportion formula.

3-Explain inclusion and exclusion criteria.

Response: Thank you for your recommendation. We have included the inclusion and exclusion criteria to this study.

Results:

1-Revision all tables.

Response: Thank you for your recommendations. We have revised the tables so that it can conform to the Plos one table formatting.

2-Table number 2 not clear. Unifying the form of writing numbers and the symbol %.

Response: Thank you for your recommendation. We have improved table 2 according to your suggestion.

---

## [Decision Letter · Decision Letter 1]

8 Apr 2020

PONE-D-19-34888R1

Mothers’ satisfaction with health extension services and the associated factors in Gamo Gofa Zone, Southern Ethiopia

PLOS ONE

Dear Mrs. Yeshitila,

Thank you for submitting your manuscript to PLOS ONE. After careful consideration, we feel that it has merit but does not fully meet PLOS ONE’s publication criteria as it currently stands. Therefore, we invite you to submit a revised version of the manuscript that addresses the points raised during the review process.

We would appreciate receiving your revised manuscript by May 23 2020 11:59PM. To enhance the reproducibility of your results, we recommend that if applicable you deposit your laboratory protocols in protocols.io, where a protocol can be assigned its own identifier (DOI) such that it can be cited independently in the future. For instructions see: http://journals.plos.org/plosone/s/submission-guidelines#loc-laboratory-protocols

We look forward to receiving your revised manuscript.

Kind regards,

Nülüfer Erbil, Ph.D, Prof.

Academic Editor

PLOS ONE

Additional Editor Comments (if provided):

Dear Author/ Authors,

The opinions and suggestions of the 3rd reviewer regarding your article are below. Submit corrections in your article according to the recommendations of the reviewer.

The opinions and suggestions of the 3rd reviewer:

Thank you for the opportunity to second time review this manuscript under title the Mothers’ satisfaction with health extension services and the associated factors in Gamo Gofa Zone, Southern Ethiopia.

Thank you for follow my recommendations. Please see care full until now some comment not corrected like the following:

1- It's better to capitalize the first letter in each word in the title.

2- At the beginning of each paragraph leaves a space and begins writing.

3- Some of grammatical errors should be corrected

4- in line 41 you are corrected the style of writing. For example, the [AOR=4.02 (95% CI 45 2.0, 8.1), You are corrected it [AOR=4.02 (95% CI 45 2.0, 8.1)],

Correct all of them in line 42, 43,44…. When you used the [ at the begin should be closed at the end. [……] . If I was right

5- Number and data of Ethics consideration letter write down under subtitle Ethics consideration.

6- Table number 2, Unifying the form of writing numbers and the symbol %

For example, you are put in some area or part the symbol of percentage and omitted in the other part.

Reviewers' comments:

Reviewer's Responses to Questions

**Comments to the Author**

1. If the authors have adequately addressed your comments raised in a previous round of review and you feel that this manuscript is now acceptable for publication, you may indicate that here to bypass the “Comments to the Author” section, enter your conflict of interest statement in the “Confidential to Editor” section, and submit your "Accept" recommendation.

Reviewer #1: All comments have been addressed

Reviewer #2: All comments have been addressed

Reviewer #3: All comments have been addressed

2. Is the manuscript technically sound, and do the data support the conclusions?

Reviewer #1: Yes

Reviewer #2: Yes

Reviewer #3: Yes

3. Has the statistical analysis been performed appropriately and rigorously? 

Reviewer #1: Yes

Reviewer #2: Yes

Reviewer #3: Yes

4. Have the authors made all data underlying the findings in their manuscript fully available?

Reviewer #1: Yes

Reviewer #2: Yes

Reviewer #3: Yes

5. Is the manuscript presented in an intelligible fashion and written in standard English?

Reviewer #1: Yes

Reviewer #2: Yes

Reviewer #3: Yes

6. Review Comments to the Author

Reviewer #1: The responses of authors to my comments were comprehensive and authors have addressed my comments comprehensively

Reviewer #2: The paper will provide good insight regarding to maternal and child health related complications secondary to dissatisfaction if publication is considered

Reviewer #3: Thank you for the opportunity to second time review this manuscript under title the Mothers’ satisfaction with health extension services and the associated factors in Gamo Gofa Zone, Southern Ethiopia.

Thank you for follow my recommendations. Please see care full until now some comment not corrected like the following:

1- It's better to capitalize the first letter in each word in the title.

2- At the beginning of each paragraph leaves a space and begins writing.

3- Some of grammatical errors should be corrected

4- in line 41 you are corrected the style of writing. For example, the [AOR=4.02 (95% CI 45 2.0, 8.1), You are corrected it [AOR=4.02 (95% CI 45 2.0, 8.1)],

Correct all of them in line 42, 43,44…. When you used the [ at the begin should be closed at the end. [……] . If I was right

5- Number and data of Ethics consideration letter write down under subtitle Ethics consideration.

6- Table number 2, Unifying the form of writing numbers and the symbol %

For example, you are put in some area or part the symbol of percentage and omitted in the other part.

7. PLOS authors have the option to publish the peer review history of their article (what does this mean?). If published, this will include your full peer review and any attached files.

Reviewer #1: Yes: Parvin Abedi

Reviewer #2: Yes: Daniel Adane Endalew

Reviewer #3: No

---

## [Author Response · Author response to Decision Letter 1]

14 Apr 2020

Authors’ response to Editor and Reviewer comments

Additional Editor Comments (if provided):

Comment: 

Dear Author/ Authors,

The opinions and suggestions of the 3rd reviewer regarding your article are below. Submit corrections in your article according to the recommendations of the reviewer.

Response: Thank you dear Prof. Nülüfer Erbil, we have given a point-by-point response to the reviewer’s comment and have made the changes in the revised manuscript shown with track changes.

xxxxxxxxxxxxxxxxxxxxxxxxxxxxxxxxxxxxxxxxxxxxxxxxxxxxxxxxxxxxxxxxxxxxxxxxxxxxxxxxxxxxxxxxxxxxxxxxxxxxxxxxxxxxxxxxxxxxxxxxxxxxxxxxxxxxxxxxxxxxxxxxxxxxxxxxxxxx

Reviewer 3 comments

General comments: Thank you for follow my recommendations. 

Response: Thank you for your inspiring comment.

Specific comments:

Please see care full until now some comment not corrected like the following.

1- It's better to capitalize the first letter in each word in the title.

Response: As we have replied in the first revision, we left it as it is just to conform to Plos One title guide which recommends the title to be in a sentence case. Kindly consider our efforts to conform to the journal formatting style. (Please refer to https://journals.plos.org/plosone/s/submission-guidelines, specifically under the section of Title page

2- At the beginning of each paragraph leaves a space and begins writing.

Response: We ask apologies for missing this comment. We have now corrected this in the revised submission. 

3- Some of grammatical errors should be corrected

Response: We are so sorry; we could not get the exact places in the document where these corrections are actually needed. However, we have gone all over the texts and corrected the errors we are able to detect.

4- in line 41 you are corrected the style of writing. For example, the [AOR=4.02 (95% CI 45 2.0, 8.1), You are corrected it [AOR=4.02 (95% CI 45 2.0, 8.1)],

Correct all of them in line 42, 43,44…. When you used the [ at the begin should be closed at the end. [……] . If I was right.

Response: Thank you very much for your meticulous observation to the errors we were unable to detect. We have corrected it and have made the changes in the “revised man¬uscript with track changes” file.

5- Number and data of Ethics consideration letter write down under subtitle Ethics consideration 

Response: We have included the number and date from the ethics assurance letter under the “Ethics consideration” section of methods.

6- Table number 2, unifying the form of writing numbers and the symbol %

For example, you are put in some area or part the symbol of percentage and omitted in the other part.

Response: We have corrected in accordance with your recommendations in Table 2.

xxxxxxxxxxxxxxxxxxxxxxxxxxxxxxxxxxxxxxxxxxxxxxxxxxxxxxxxxxxxxxxxxxxxxxxxxxxxxxxxxxxxxxxxxxxxxxxxxxxxxxxxxxxxxxxxxxxxxxxxxxxxxxxxxxxxxxxxxxxxxxxxxxxxxxxxxxxx

---

## [Editor Report · Decision Letter 2]

24 Apr 2020

Mothers’ satisfaction with health extension services and the associated factors in Gamo Goffa Zone, Southern Ethiopia

PONE-D-19-34888R2

Dear Dr. Yeshitila,

We are pleased to inform you that your manuscript has been judged scientifically suitable for publication and will be formally accepted for publication once it complies with all outstanding technical requirements.

With kind regards,

Nülüfer Erbil, Ph.D, Prof.

Academic Editor

PLOS ONE
---

## [Editor Report · Acceptance letter]

29 Apr 2020

PONE-D-19-34888R2 

Mothers’ satisfaction with health extension services and the associated factors in Gamo Goffa zone, Southern Ethiopia 

Dear Dr. Yeshitila:

I am pleased to inform you that your manuscript has been deemed suitable for publication in PLOS ONE. Congratulations! Your manuscript is now with our production department. 

With kind regards,

on behalf of

Mrs. Nülüfer Erbil 

Academic Editor

PLOS ONE